

# Genome-wide identification and expression analysis of the MYB transcription factor in moso bamboo (*Phyllostachys edulis*)

Kebin Yang[1,2], Ying Li[1,2], Sining Wang[1,2], Xiurong Xu[1,2], Huayu Sun[1,2], Hansheng Zhao[1,2], Xueping Li[1,2] and Zhimin Gao[1,2]

[1] Institute of Gene Science for Bamboo and Rattan Resources, International Center for Bamboo and Rattan, Beijing, China
[2] State Forestry Administration Key Open Laboratory on the Science and Technology of Bamboo and Rattan, Beijing, China

## ABSTRACT

The MYB family, one of the largest transcription factor (TF) families in the plant kingdom, plays vital roles in cell formation, morphogenesis and signal transduction, as well as responses to biotic and abiotic stresses. However, the underlying function of bamboo MYB TFs remains unclear. To gain insight into the status of these proteins, a total of 85 PeMYBs, which were further divided into 11 subgroups, were identified in moso bamboo (*Phyllostachys edulis*) by using a genome-wide search strategy. Gene structure analysis showed that *PeMYB*s were significantly different, with exon numbers varying from 4 to 13. Phylogenetic analysis indicated that PeMYBs clustered into 27 clades, of which the function of 18 clades has been predicted. In addition, almost all of the *PeMYB*s were differently expressed in leaves, panicles, rhizomes and shoots based on RNA-seq data. Furthermore, qRT-PCR analysis showed that 12 *PeMYB*s related to the biosynthesis and deposition of the secondary cell wall (SCW) were constitutively expressed, and their transcript abundance levels have changed significantly with increasing height of the bamboo shoots, for which the degree of lignification continuously increased. This result indicated that these *PeMYB*s might play fundamental roles in SCW thickening and bamboo shoot lignification. The present comprehensive and systematic study on the members of the MYB family provided a reference and solid foundation for further functional analysis of MYB TFs in moso bamboo.

Corresponding author
Zhimin Gao, gaozhimin@icbr.ac.cn

## INTRODUCTION

Secondary cell wall (SCW) deposition and lignification is one of the most important and valuable biological activities for plant growth and development, and the SCW is one of most abundant raw materials on earth and has a wide range of industrial applications (*Oh, Park & Han, 2003*). Lignified SCWs contribute to the excellent material quality of wood species and lignocellulosic biomass as the most environmentally cost-effective

renewable sources of energy (*Seth, 2003*; *Pauly & Keegstra, 2010*; *Cassan-Wang et al., 2013*). In the context of the economic and environmental significance of SCW, the SCWs of plant cells have received increasing attention (*Carroll & Somerville, 2009*; *Pauly & Keegstra, 2010*; *Cassan-Wang et al., 2013*). Great progress had been made in the understanding of the genetic regulation of SCW biosynthesis, which includes several consecutive processes, mainly at the transcriptional level (*Hirano et al., 2013*). The regulation of SCW biosynthesis involves a complex network that includes transcription factors (TFs) and miRNAs, among which most of the TFs belonging to the MYB family have been reported to function as links between upstream NAC TFs and downstream structural genes (*Cassan-Wang et al., 2013*; *Hussey et al., 2013*; *Nakano et al., 2015*).

The MYB family is one of the largest TF families, and it is named after a highly conservative sequence (MYB DNA-binding domain) located at the N-terminus of these proteins. Each MYB DNA-binding domain comprises 1–4 serial and nonredundant imperfect repeats (R1, R2, R3 and R4). Each repeat contains approximately 50–53 amino acids that form three $\alpha$-helices, and a helix–turn–helix (HTH) structure is formed between the second and third $\alpha$-helices (*Lipsick, 1996*; *Stracke, Werber & Weisshaar, 2001*). Diametrically, the amino acid sequence outside of the DNA-binding domain, the C-terminus, is an activated structure, that is, highly divergent in length and sequence, which gives rise to the functional diversity of the MYB proteins (*Kranz, Scholz & Weisshaar, 2000*; *Jin & Martin, 1999*). Based on the number of repeats, the MYB family is classified into four subfamilies, namely, 1R-MYB (MYB-related), 2R-MYB (R2R3-MYB), 3R-MYB (R1R2R3-MYB) and 4R-MYB (*Dubos et al., 2010*; *He et al., 2016*). In plants, R2R3-MYB are the largest and most common MYB TFs (*Dubos et al., 2010*; *Du et al., 2013*; *Niu, Jiang & Xu, 2016*), and these proteins are involved in almost all aspects and stages of plant growth and development. At present, functional studies on MYB have been mainly focused on R2R3-MYB TFs and have rarely addressed the other three TF subfamilies.

To date, many members of the MYB TF family have been recognized and studied in many crops and horticultural plants, such as *Arabidopsis*, poplar, maize, soybean, pineapple, upland cotton and beet (*Wilkins et al., 2009*; *Dubos et al., 2010*; *Du et al., 2012a, 2012b*; *Stracke et al., 2014*; *Salih et al., 2016*; *Liu et al., 2017*). MYB TFs are involved in regulating the growth and development of various plants by participating in many physiological and biochemical processes, such as cell and petal morphogenesis and flavonol biosynthesis (*Baumann et al., 2007*; *Stracke et al., 2007*), and one of the most important functions of these proteins is regulating the synthesis of SCWs (*Yang & Wang, 2016*). Importantly, MYB TFs play central roles in the transcriptional regulation of the deposition of the plant SCW. Many MYB genes have been identified as key genes involved in SCW synthesis, such as *AtMYB58*, *AtMYB63*, *AtMYB46*, *AtMYB83* and *AtMYB103* (*Zhou et al., 2009*; *McCarthy, Zhong & Ye, 2009*; *Hussey et al., 2011*; *Guo et al., 2017*). In contrast, three other MYB genes, *AtMYB4*, *AtMYB7* and *AtMYB32*, can inhibit the expression of NAC genes, supporting the idea that these genes are negative regulators of SCW synthesis (*Zhong et al., 2008*). Similarly, homologous genes have also

been found in other species. *BplMYB46* of *Betula platyphylla* and *SbMYB60* of *Sorghum bicolor* are involved in the deposition of SCW through regulating lignin synthesis (*Zhou et al., 2009*; *Scully et al., 2016*; *Guo et al., 2017*). However, the function of MYBs related to SCW synthesis in bamboo is still unknown.

Bamboo is characterized by fast growth with a long vegetative period and high yield, which have high value in various industries, such as papermaking, forestry and crafts. Furthermore, young bamboo shoots can be used for food (*Wu et al., 2015*). Bamboo is also regarded as an emerging and important sources of lignocellulosic biomass energy. The rapid growth of bamboo is accompanied with SCW thickening and lignification, which plays a vital role in the improvement of excellent wood property for broad application in the manufacturing industry (*Yu, 2003*; *Gao et al., 2010*). MYB TFs are involved in SCW development by regulating the expression of lignin, cellulose and hemicellulose synthesis-related genes and therefore indirectly affect material properties (*Hussey et al., 2013*; *Soler et al., 2015*). In addition, the number of MYB TF family members greatly varies from species to species, and even homologous genes differ in gene structure and function among different species. Therefore, it is necessary to explore the specific structural and functional characteristics of bamboo MYB genes for further study. The goal of this study is to fully understand the status of MYB TFs related to secondary wall synthesis in moso bamboo.

Moso bamboo (*Phyllostachys edulis*) is an important woody bamboo with high value for lignocellulosic biomass and is the only bamboo that has been sequenced. In moso bamboo, we hypothesize that (i) the members of the MYB TF family might have similar gene structures and different numbers of family members compared to those in *Arabidopsis thaliana*; (ii) the expression pattern of MYB genes may show significant tissue specificity; (iii) and the MYB genes would primarily be involved in the biosynthesis and deposition of the SCW. To test these hypotheses, we identified MYB genes in the whole genome of moso bamboo and investigated their gene structural characteristics and their evolutionary relationships. We further investigated the tissue-specific expression patterns of these genes by using RNA-seq data. Finally, we validated the role of MYB genes related to the biosynthesis of SCW by qRT-PCR using bamboo shoots with different lignification degrees. Thus, the present study provided a starting point for further functional analysis of MYB genes in moso bamboo, and laid the foundation of selecting candidate genes for genetic engineering in bamboo breeding.

## MATERIALS AND METHODS

### Plant material

To examine expression differences of the MYB genes involved in the biosynthesis of SCWs during the lignification of shoots in moso bamboo, the basal parts of bamboo shoots with different heights (0.2, 1.0, 3.0 and 6.7 m) were collected from the bamboo forest experimental site of Jiangxi Academy of Forestry located in Nanchang City, Jiangxi Province, China (E115°46′1″; N28°45′57″). Shoots of 0.2, 1.0, 3.0 and 6.7 m in height correspondingly belonged to preliminary, ascending, prosperous and late shoot developmental stages, in which the degree of lignification gradually increased.

These mixed samples from three individual shoots with different heights were immediately frozen in liquid nitrogen and stored at −80 °C until further use. Meanwhile, a part of the same samples had been fixed in formalin-acetic acid-alcohol (FAA) and kept in a refrigerator at 4 °C before use.

## Histological methods

Shoot samples were taken out from FAA fixative, dehydrated through a graded series of polyethylene glycol (PEG) (Tianjin Guangfu Fine Chemical Research Institute, Tianjin, China) at 80 °C, first with the equal volume mixture of PEG 1000 and deionized water, followed by the equal volume mixture of PEG 1000 and PEG 4000, and finally with PEG 4000. The samples embedded in PEG 4000 at room temperature were used for section making with a rotary microtome (Leica RM2165; Leica, Frankfurt, Germany). Tissue sections (10 μm) were cut transversely from the embedded samples and gently transferred to clean slides with brushes. The sections were fully expanded and stained with toluidine blue (*Windham et al., 2018*), which were heated at 80 °C for 3 min. The slides were washed with deionized water and dried. Cover slips were cemented over stained sections and viewed with an Olympus CX31 microscope.

## Database search for MYBs in moso bamboo

The genome of moso bamboo (*Peng et al., 2013b*) and the BambooGDB database (http://www.bamboogdb.org/) (*Zhao et al., 2014*) facilitated a genome-wide analysis of the bamboo gene families (*Sun et al., 2016*; *Huang et al., 2016*). To identify the potential members of the MYB TF family in moso bamboo, we performed multiple sequence blast and alignment. First, the putative MYB sequences were downloaded from BambooGDB. Then, the MYB conserved domains in putative sequences were examined by using BLASTN and BLASTP. Finally, all MYB sequences were manually inspected to ensure that the putative protein models contained two, three and four MYB repeats, and the protein sequences that did not contain conserved domains were deleted. The MYB TFs were named according to their BambooGDB assembly names.

The basic characteristics of the potential MYB TF members in moso bamboo were further analyzed, including the predicted proteins and the physicochemical parameters. The predicted molecular weights (MWs) and isoelectric points (pIs) of the MYB proteins were analyzed using ProtParam (http://web.expasy.org/protparam/).

## WebLogo and gene structure analysis

To reveal the sequence features of the conserved DNA-binding domains in R2R3-MYB proteins, the sequences of the R2 and R3 MYB repeats in all PeR2R3-MYB proteins were compared using the ClustalW program in MEGA (version 6.0) (*Tamura et al., 2013*). The same method was used to perform the multiple sequence alignment encompassing 82 R2R3-MYB proteins from moso bamboo, 126 from *A. thaliana* and 111 from *Oryza sativa* (rice). The distribution of the amino acid residues at the corresponding positions in the conserved domains of R2R3-PeMYBs were generated using the WebLogo program with default parameters (http://weblogo.berkeley.edu/logo.cgi) (*Crooks et al., 2004*).

For the gene structure analysis, the exons and introns of the MYB genes (*PeMYBs*) were illustrated using the Gene Structure Display Server (GSDS; http://gsds.cbi.pku.edu.cn/) (*Guo et al., 2007*) to align the cDNA sequences with the corresponding genomic DNA sequences from the BambooGDB.

## Phylogenetic analysis and function prediction

To explore the evolutionary relationships among MYBs in moso bamboo and predict the functions of these MYBs, the MYB sequences of *Arabidopsis* and rice were downloaded from the *Arabidopsis* genome TAIR (The *Arabidopsis* Information Resource) release 10.0 (http://www.arabidopsis.org/) and the rice genome annotation database (http://rice.plantbiology.msu.edu/index.shtml, release 7.0). A neighbor-joining (NJ) phylogenetic tree was constructed with ClustalW to align the full-length of MYB amino acid sequences (85 PeMYBs and 132 AtMYBs) using MEGA (version 6.0) with the following parameters: Poisson correction, pairwise deletion, and bootstrap analysis with 1,000 replicates. The PeMYBs were classified according to their phylogenetic relationships with the corresponding 27 clades of AtMYBs (*Stracke, Werber & Weisshaar, 2001*; *Dubos et al., 2010*). Additionally, the biological functions of PeMYBs were predicted according to the aforementioned phylogenetic tree and previously studies homologous *Arabidopsis* proteins with validated specific function (*Zhong et al., 2008*; *McCarthy, Zhong & Ye, 2009*; *Dubos et al., 2010*; *Li et al., 2016b*).

## Tissue specific expression analysis of MYB genes

To study the expression patterns of MYB genes in different tissues of moso bamboo, the transcriptome data for the leaves, panicles, roots, rhizomes and shoots at different developmental stages were downloaded from the Short Read Archive of NCBI. The gene expression abundance was calculated by the reads per kilobase per million mapped reads value of each MYB gene. For the convenience of statistics, logarithm (Log) was used for each expression as base 2. The heat map of gene expression was decorated using Matrix2png (http://www.chibi.ubc.ca/matrix2png/).

## Real-time PCR analysis

Total RNA was extracted using the plant RNA extraction kit (Qiagen, Dusseldorf, Germany) according to the manufacturer's instructions. The integrity of the total RNA was verified through agarose gel electrophoresis, and the purity and concentration of the total RNA was determined by spectrophotometry. The first strand cDNA was synthesized by a reverse transcription system (Promega, Madison, WI, USA). For each 20-μL reaction, 1,000 ng of total RNA was used, and the synthesis was performed at the 42 °C for 45 min. The final cDNA product was diluted fivefold prior to use.

*PeMYBs* involved in SCW synthesis were screened according to their phylogenetic relationships with the corresponding *AtMYBs*. Based on the multiple alignments, 12 specific primers for different *PeMYBs* were designed by Primer Premier 5.0 software and empirically adjusted for gene expression analysis (Table S1). Additionally, all primers

showing a clear specific melting peak by real-time melting curve analysis, consistent with the results of agarose gel electrophoresis for specific PCR products, were used for further analysis. The qRT-PCR was performed with the Roche Light Cycler 480 SYBR Green 1 Master kit on a qTOWER2.2 system (Analytik Jena, Jena, Germany). The qRT-PCR program involved 95 °C for 10 min, followed by 40 cycles at 95 °C for 10 s and 60 °C for 10 s. The 10.0 μL reaction volume contained 5.0 μL of 2× SYBR Green 1 Master Mix, 0.8 μL of cDNA, 0.1 μL of primer (10.0 mM, each) and 4.0 μL of ddH$_2$O. *NTB* and *TIP41* were used as reference genes (*Fan et al., 2013*). The $2^{-\Delta\Delta Ct}$ method was used for the analysis and visualization of real-time PCR data generated by multiple technical replicates (*Liu et al., 2013*).

## Statistical analysis

Analyses were performed with SPSS Statistics for Windows (Version 22.0; SPSS Inc., Chicago, IL, USA). All data were the average and standard error of three biological replicates. One-way analysis of variance was used to evaluate the statistical significance of differences among means using SPSS software. Single and double asterisks indicate significant differences at the levels of $p < 0.05$ and $p < 0.01$, respectively.

# RESULTS

## Identification, protein characteristics and conserved DNA-binding domain analysis of MYB TFs in moso bamboo

Through comprehensive comparison analysis, we identified a set of 85 MYB proteins containing MYB DNA-binding domains in moso bamboo (Table S2), which included 82 typical R2R3-MYB proteins (2R-MYB), two R1R2R3-MYB proteins (3R-MYB) and one 4R-like MYB protein (4R-MYB). According to the numbering order of MYB in BambooGDB, 2R-MYB proteins were named PeMYB1–PeMYB82, while 3R-MYB and 4R-MYB proteins were named PeMYB3R-1–PeMYB3R-2 and PeMYB4R-1, respectively. As shown in Table 1, the length of the corresponding estimated polypeptides ranged from 199 to 1,024 amino acids, the calculated MW of PeMYBs ranged from 22.3 to 112.4 kDa, and the calculated theoretical pI of PeMYBs was from 5.15 to 11.67. The majority of R2R3-MYB proteins were approximately 300 amino acids with MWs of approximately 30 kDa. However, 82 R2R3-MYB proteins presented irregular characteristics of theoretical pI, leading to 50 acid proteins and 32 basic proteins. Among the 85 PeMYBs, PeMYB3R-2 was the longest protein with 1,024 amino acids, while the shortest protein was PeMYB49 with 199 amino acids.

To further investigate and identify the characteristics of homologous domains in R2R3-MYB proteins, multiple sequence alignment and WebLogo were performed using the amino acid sequences of R2 and R3 repeats in 82 PeMYBs (R2R3-MYB). As shown in Fig. 1 and Fig. S1, R2R3-MYB proteins contained R2 and R3 repeats, suggesting that the characterized PeMYBs were similar to those of other species, with basic R2 and R3 structures of [-W-(X19)-W-(X19)-W-] and [-F-(X18)-W-(X18)-W-], respectively. The results showed that each repeat included a highly conserved triplet of tryptophan (W)

**Table 1 Nomenclature and protein information of MYBs in moso bamboo.**

| Nomenclature used for this paper | Bamboo GDB assembly name | Pl (aa) | MW (Da) | pI | Nomenclature used for this paper | Bamboo GDB assembly name | Pl (aa) | MW (Da) | pI |
|---|---|---|---|---|---|---|---|---|---|
| PeMYB1 | PH01000001G2130 | 277 | 30,332.85 | 7.08 | PeMYB44 | PH01000847G0490 | 275 | 30,661.58 | 5.38 |
| PeMYB2 | PH01000005G1390 | 286 | 31,377.37 | 6.24 | PeMYB45 | PH01000912G0430 | 247 | 27,623.01 | 5.68 |
| PeMYB3 | PH01000006G2680 | 378 | 40,189.57 | 5.94 | PeMYB46 | PH01000958G0180 | 988 | 111,211.3 | 5.15 |
| PeMYB4 | PH01000008G0500 | 288 | 30,732.33 | 7.02 | PeMYB47 | PH01001022G0490 | 254 | 27,279.34 | 8.30 |
| PeMYB5 | PH01000008G3080 | 308 | 34,411.17 | 5.43 | PeMYB48 | PH01001064G0370 | 392 | 43,742.65 | 5.75 |
| PeMYB6 | PH01000009G0060 | 554 | 59,864.36 | 4.95 | PeMYB49 | PH01001084G0440 | 199 | 22,335.16 | 9.78 |
| PeMYB7 | PH01000014G1850 | 257 | 27,923.20 | 7.54 | PeMYB50 | PH01001133G0430 | 228 | 26,192.18 | 8.83 |
| PeMYB8 | PH01000028G0940 | 284 | 30,699.39 | 6.18 | PeMYB51 | PH01001174G0490 | 441 | 49,726.99 | 6.06 |
| PeMYB9 | PH01000029G1950 | 587 | 63,493.82 | 6.17 | PeMYB52 | PH01001208G0070 | 204 | 23,122.78 | 6.31 |
| PeMYB10 | PH01000030G0050 | 363 | 39,417.23 | 5.03 | PeMYB53 | PH01001287G0090 | 253 | 28,133.68 | 6.77 |
| PeMYB11 | PH01000041G2150 | 332 | 35,798.18 | 6.87 | PeMYB54 | PH01001342G0270 | 272 | 30,538.57 | 10.6 |
| PeMYB12 | PH01000043G2100 | 232 | 25,140.23 | 9.01 | PeMYB55 | PH01001430G0250 | 262 | 28,784.47 | 7.60 |
| PeMYB13 | PH01000053G1340 | 309 | 34,152.74 | 6.55 | PeMYB56 | PH01001622G0290 | 318 | 33,660.00 | 9.13 |
| PeMYB14 | PH01000060G0800 | 276 | 30,672.17 | 5.90 | PeMYB57 | PH01001925G0330 | 247 | 27,606.01 | 7.72 |
| PeMYB15 | PH01000064G1730 | 300 | 32,262.85 | 7.01 | PeMYB58 | PH01001991G0310 | 316 | 36,090.86 | 8.52 |
| PeMYB16 | PH01000066G1200 | 273 | 29,409.31 | 9.26 | PeMYB59 | PH01002000G0040 | 235 | 25,734.12 | 8.93 |
| PeMYB17 | PH01000068G1000 | 337 | 36,928.09 | 8.46 | PeMYB60 | PH01002082G0250 | 284 | 30,453.86 | 6.40 |
| PeMYB18 | PH01000068G1470 | 308 | 32,931.66 | 6.96 | PeMYB61 | PH01002092G0300 | 386 | 42,365.66 | 5.04 |
| PeMYB19 | PH01000177G0890 | 304 | 32,461.31 | 9.46 | PeMYB62 | PH01002104G0150 | 278 | 30,138.61 | 9.81 |
| PeMYB20 | PH01000198G1320 | 313 | 34,110.22 | 7.66 | PeMYB63 | PH01002139G0430 | 289 | 32,202.79 | 8.33 |
| PeMYB21 | PH01000209G0490 | 316 | 34,250.01 | 7.84 | PeMYB64 | PH01002184G0220 | 325 | 34,980.08 | 4.63 |
| PeMYB22 | PH01000212G0840 | 424 | 46,705.17 | 6.32 | PeMYB65 | PH01002276G0160 | 328 | 35,201.40 | 6.87 |
| PeMYB23 | PH01000234G0090 | 328 | 35,312.59 | 6.68 | PeMYB66 | PH01002680G0080 | 273 | 30,149.20 | 5.81 |
| PeMYB24 | PH01000302G0910 | 295 | 32,877.25 | 5.76 | PeMYB67 | PH01002704G0220 | 313 | 35,029.41 | 6.38 |
| PeMYB25 | PH01000305G0990 | 263 | 29,168.77 | 9.74 | PeMYB68 | PH01002707G0220 | 304 | 33,482.32 | 6.10 |
| PeMYB26 | PH01000332G0140 | 240 | 27,343.46 | 8.85 | PeMYB69 | PH01002736G0020 | 426 | 47,126.77 | 6.17 |
| PeMYB27 | PH01000345G0740 | 266 | 29,911.06 | 5.90 | PeMYB70 | PH01002868G0200 | 272 | 29,620.53 | 9.41 |
| PeMYB28 | PH01000374G0780 | 509 | 56,256.12 | 8.06 | PeMYB71 | PH01003180G0140 | 327 | 35,295.85 | 6.06 |
| PeMYB29 | PH01000386G0660 | 346 | 37,877.37 | 5.23 | PeMYB72 | PH01003507G0040 | 294 | 31,738.31 | 5.44 |
| PeMYB30 | PH01000392G0510 | 339 | 36,821.16 | 5.63 | PeMYB73 | PH01003809G0130 | 341 | 37,143.94 | 5.15 |
| PeMYB31 | PH01000415G0010 | 292 | 31,593.87 | 7.68 | PeMYB74 | PH01003918G0100 | 445 | 49,270.93 | 5.90 |
| PeMYB32 | PH01000415G0090 | 284 | 30,871.39 | 6.31 | PeMYB75 | PH01004818G0120 | 272 | 30,731.28 | 6.40 |
| PeMYB33 | PH01000427G0040 | 261 | 28,722.34 | 7.03 | PeMYB76 | PH01004865G0070 | 240 | 27,157.40 | 11.67 |
| PeMYB34 | PH01000445G0700 | 310 | 34,607.94 | 5.64 | PeMYB77 | PH01005192G0010 | 289 | 30,825.37 | 6.35 |
| PeMYB35 | PH01000462G0290 | 322 | 35,252.55 | 5.59 | PeMYB78 | PH01005460G0120 | 297 | 32,145.28 | 8.11 |
| PeMYB36 | PH01000501G0490 | 321 | 34,627.14 | 7.69 | PeMYB79 | PH01005515G0070 | 357 | 39,436.22 | 6.51 |
| PeMYB37 | PH01000508G0100 | 334 | 36,499.74 | 5.85 | PeMYB80 | PH01005685G0080 | 338 | 36,493.24 | 6.06 |
| PeMYB38 | PH01000515G0560 | 276 | 31,171.97 | 5.61 | PeMYB81 | PH01005828G0060 | 317 | 35,395.00 | 5.40 |
| PeMYB39 | PH01000569G0800 | 264 | 28,563.96 | 5.50 | PeMYB82 | PH01007341G0010 | 272 | 29,204.97 | 9.20 |
| PeMYB40 | PH01000595G0350 | 427 | 47,009.56 | 6.23 | PeMYB3R-1 | PH01000054G0640 | 817 | 90,393.02 | 9.36 |

(Continued)

| Nomenclature used for this paper | Bamboo GDB assembly name | Pl (aa) | MW (Da) | pI | Nomenclature used for this paper | Bamboo GDB assembly name | Pl (aa) | MW (Da) | pI |
|---|---|---|---|---|---|---|---|---|---|
| PeMYB41 | PH01000604G0860 | 236 | 26,573.87 | 7.71 | PeMYB3R-2 | PH01000812G0550 | 1,024 | 112,448.60 | 5.22 |
| PeMYB42 | PH01000617G0820 | 275 | 30,759.97 | 5.50 | PeMYB4R-1 | PH01002148G0200 | 930 | 102,980.90 | 9.23 |
| PeMYB43 | PH01000729G0470 | 218 | 23,255.01 | 8.25 | | | | | |

**Note:**
Pl, polypeptide length; MW, molecular weight; Da, Dalton; pI, isoelectric point; aa, amino acids.

residues, and each characteristic of W was separated by 18 or 19 amino acids, which were located at positions 6, 26 and 46 of the R2 repeat and 58, 77 and 96 of the R3 repeat (Fig. 1; Fig. S1). In addition to the highly conserved W, glutamic (E)-10, aspartic (D)-11 cysteine (C)-42, arginine (R)-45 in the R2 repeat, leucine (L)-50 in the linker region, and arginine (R)-87, threonine (T)-88 in the R3 repeat were also completely conserved (Fig. 1; Fig. S2). Interestingly, the first conservative tryptophan (located at 58) in R3 was mostly substituted by phenylalanine (F), marginally by isoleucine (I) and leucine (L), and the individual tryptophan was conserved. Furthermore, the glycine (G) (located at 74) was replaced by alanine (A) for PeMYB73 and PeMYB30, and threonine (T) for PeMYB60, respectively. As shown in Fig. 1 and Fig. S1, the conservative areas in the MYB DNA-binding domain were mainly located between the second and third W of the two R repeats (the third helix of the HTH domain in each repeat). However, the amino acid sequence between the first and second W of each R repeat in the MYB DNA-binding domain was relatively unconserved.

## Phylogenetic analysis of MYB TFs based on gene structure

There was a significant difference and diversity in the gene structure among the members of *PeMYBs*, including the number and relative location of exons and introns. As shown in Fig. 2, *PeMYB56* and *PeMYB78* had no introns, while the number of introns in the other members varied from 1 to 13. According to their predicted structures, the majority of *PeMYBs* contained two or three exons, and the number of members with the above characteristics was 17 and 53, respectively. To reveal the phylogenetic relationship of the MYB proteins, we performed multiple sequence alignment using the amino acid sequences of 85 PeMYBs. According to the similarity and systematic evolution of the sequences, PeMYBs were divided into 11 subgroups (designated as S1–S11) and two low homology MYB proteins (PeMYB46 and PeMYB4R-1), and each subgroup had 4–13 members. The most highly homologous members in the same subgroup generally shared the same or parallel exon/intron patterns, showing similar quantity, location and exon length. For instance, four *PeMYBs* (*PeMYB26*, *PeMYB76*, *PeMYB16* and *PeMYB59*) in S5 included two exons and one intron. Notably, one or more pairs of PeMYBs with highly homology were found in the terminal nodes of each subgroup, suggesting that these proteins share similar functions. There was also an exception of S9, in which the location and length of exons and introns were significantly different with low genetic similarity among the members.

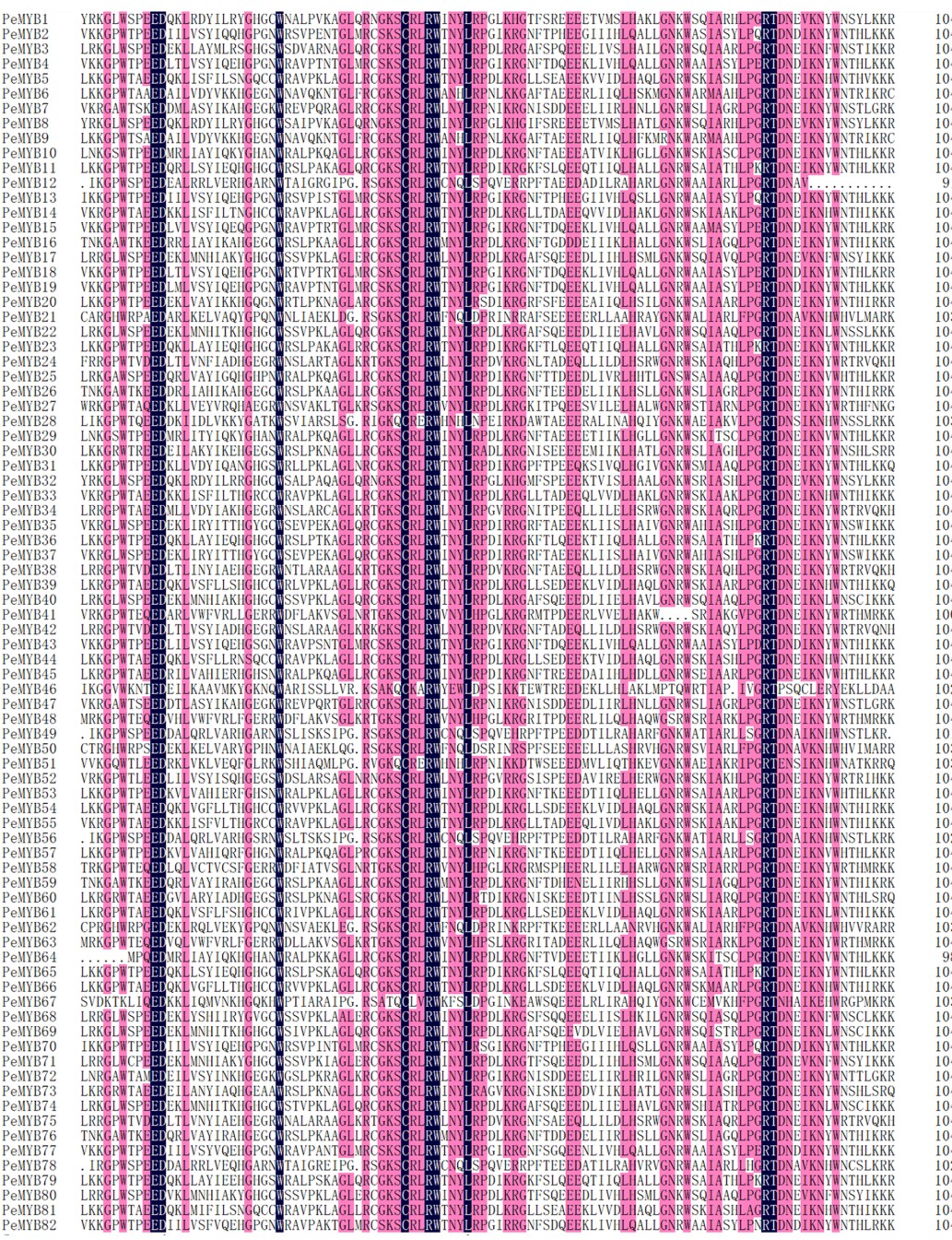

**Figure 1 Multiple alignment of the amino acid sequences of 82 moso bamboo R2R3-MYB domains.**

## Putative functions of PeMYBs in moso bamboo

MYB TFs in *A. thaliana* are divided into 27 clades, and the function of each clade had been annotated (*Zhong et al., 2008*; *McCarthy, Zhong & Ye, 2009*; *Dubos et al., 2010*; *Li et al., 2016b*). It is assumed that homologous proteins that clustered together typically have similar functions, suggesting that the PeMYBs had similar functions as AtMYBs in the same clade. Therefore, the functions of PeMYBs were predicted and summarized by

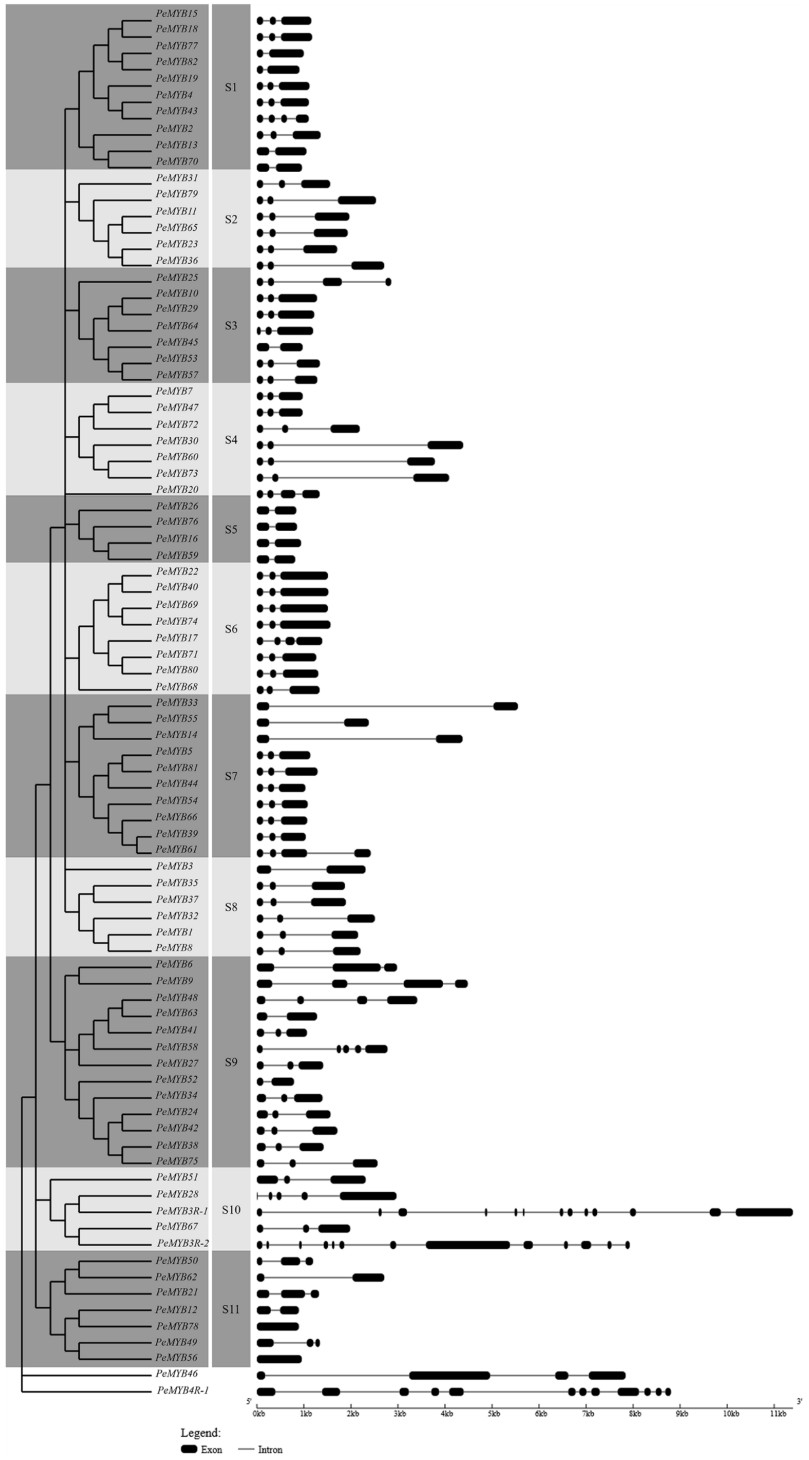

**Figure 2 Phylogenetic relationships and gene structures of MYB genes in moso bamboo.** The amino acid sequences of 85 PeMYBs were aligned by the Clustal W program in MEGA, and the phylogenetic tree was constructed by the NJ method with 1,000 bootstrap replicates. Bootstrap values >50 were indicated on the nodes. Different subgroups were marked with alternating tones of a gray background to make subgroups identification easier. Exon/intron structures of the *PeMYBs*: black boxes represented exons and spaces between the black boxes correspond to introns.

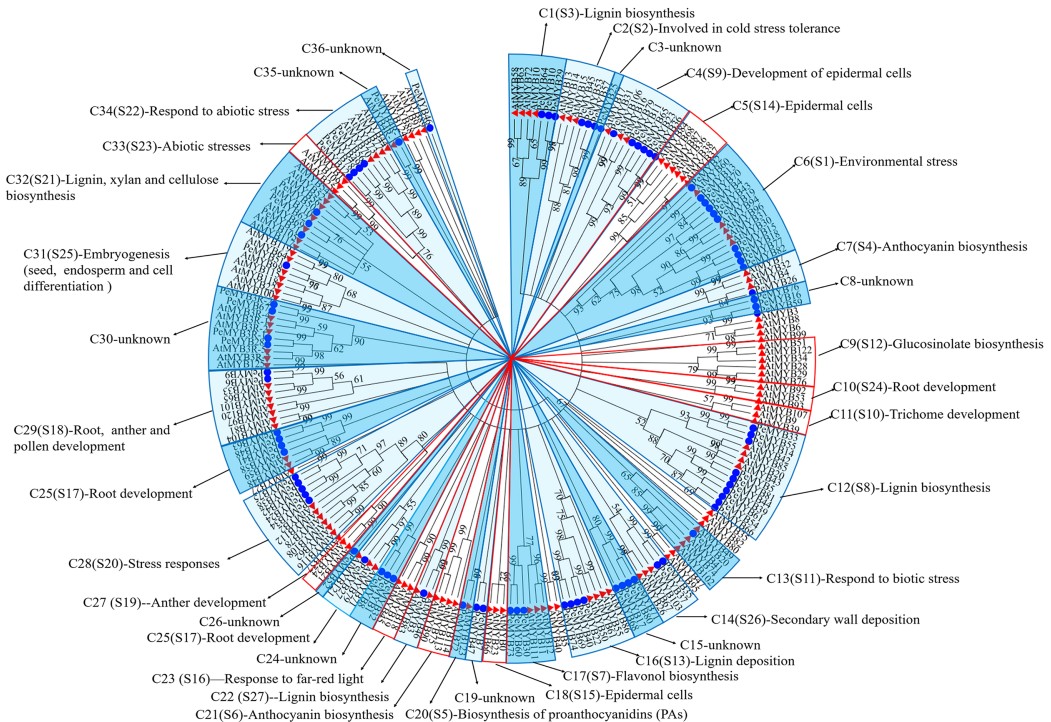

**Figure 3** Putative functions of the MYB proteins in moso bamboo based on the phylogenetic tree along with MYBs from Arabidopsis. The circular unrooted tree was generated by NJ method with 1,000 bootstrap replicates. Different subclades were marked with alternating tones of a blue background. The red boxes indicated that species-specific subclades of Arabidopsis.

comparison with those of AtMYBs (Fig. 3; Table S3). An NJ unrooted phylogenetic tree was constructed using 85 PeMYBs and 132 AtMYBs (Fig. 3). The results showed that all MYB members from the two species were clustered into 36 clades (designated as C1–C36), including 20 clades common to the two species, and seven and nine species-specific clades of moso bamboo and *Arabidopsis*, respectively.

According to the above analysis, 65 PeMYBs belonging to 18 function-annotated clades, and 20 PeMYBs belonging to nine function-unknown clades were found. Based on the annotation, 65 PeMYBs were divided into four functional classes. Class I, including six clades (C1, C12, C14, C16, C22 and C32), was responsible for SCW formation by regulating the biosynthesis and deposition of lignin, cellulose and hemicellulose. Class II, including five clades (C2, C6, C13, C28 and C34), was involved in responses to biotic and abiotic stresses by regulating the ABA pathway. Class III, including 5 clades (C4, C20, C25, C29 and C31), played important roles in morphogenesis and organogenesis, such as root, epidermal cell, anther, vegetative and stomatal cell development and embryogenesis. Class IV, including 2 clades (C7 and C17), was involved in regulating secondary metabolism, such as anthocyanins and flavonols biosynthesis. Class IV, including two clades (C4 and C31), was involved in regulating secondary metabolism, such as anthocyanins and flavonols biosynthesis. Thus, these results suggested that PeMYBs have a wide range of functions and may play important roles in the growth and development of moso bamboo.

## Tissue-specific expression analysis of *PeMYB*s by using the transcriptome data of moso bamboo

The tissue-specific expression of *PeMYB*s was analyzed by constructing a heat map using the transcriptome data of moso bamboo (*Peng et al., 2013b*). The results showed significant differences in the expression profiles of *PeMYB*s in different tissues, and most of the *PeMYB*s showed significant tissue specificity. The expression of all 85 *PeMYB*s was detected in at least one tissue, and 52 *PeMYB*s were expressed in all tissues, with transcript abundances varying from 0 to 110.12 (Fig. 4). Furthermore, some *PeMYB*s showed high expression in a particular tissue, for example, *PeMYB*s belonging to S1, S2 and S7 mostly showed dominant expression patterns in the leaves and panicles, and relatively low expression in the other four tissues. The detailed analysis of the expression patterns of *PeMYB*s showed that eight *PeMYB*s (*PeMYB12*, *PeMYB16*, *PeMYB26*, *PeMYB32*, *PeMYB46*, *PeMYB67*, *PeMYB69* and *PeMYB78*) showed dominant expression in almost all tested tissues. Interestingly, the expression of each gene in shoots was relatively lower, and nine *PeMYB*s were not detected in bamboo shoots (Fig. 4). In addition, *PeMYB*s belonging to S5 and S6 showed high expression in all tissues, whereas *PeMYB*s belonging to S9 showed low expression in all tissues. Moreover, *PeMYB62* was exclusively expressed in advanced panicle samples.

## Validation of *PeMYB*s by using real-time PCR

The function prediction results indicated that many *PeMYB*s belonged to clades related to the biosynthesis and deposition of SCW. Thus, 12 *PeMYB*s, including *PeMYB10*, *PeMYB29* and *PeMYB22* in C1, *PeMYB26* in C7, *PeMYB14* and *PeMYB33* in C12, *PeMYB37* in C14, *PeMYB22*, *PeMYB40* and *PeMYB74* in C16, *PeMYB3* in C22 and *PeMYB50* in C32, were selected for further validation. The expression profiles of the selected *PeMYB*s in moso bamboo shoots of different heights were examined by using qRT-PCR with *PeTIP41* as the reference gene.

The results showed that all 12 *PeMYB*s have changed significantly with three expression patterns: a continuously increasing trend, an increasing trend with a final decrease and a trend of slightly stable after a sharp drop with increasing bamboo shoot height. As shown in Figs. 5E and 5G, two *PeMYB*s (*PeMYB26* and *PeMYB33*) were significantly upregulated, and their relative expression were upregulated more than 70 times in 6.7 m shoots compared with that in 0.2 m shoots, particularly the expression of *PeMYB33* was the most significant at 1,810 times. In addition, the expression of *PeMYB40* was found specially, which had a totally different trend with a sharp drop in 1.0 m shoots compared to that in 0.2 m shoots, and then kept at a relatively stable low level.

However, nine *PeMYB*s (*PeMYB3*, *PeMYB10*, *PeMYB14*, *PeMYB22*, *PeMYB29*, *PeMYB37*, *PeMYB50*, *PeMYB64* and *PeMYB74*) exhibited a similar increasing expression trend, which first showed a gradual increase with a peak in 3.0 m shoots, and then a decrease in 6.7 m shoots (Figs. 5A–5D, 5F and 5H–5L). Except *PeMYB3*, all the other eight *PeMYB*s were significantly upregulated more than two times in 3.0 m shoots compared with those in 1.0 m shoots, and the highest one was *PeMYB74* with more than

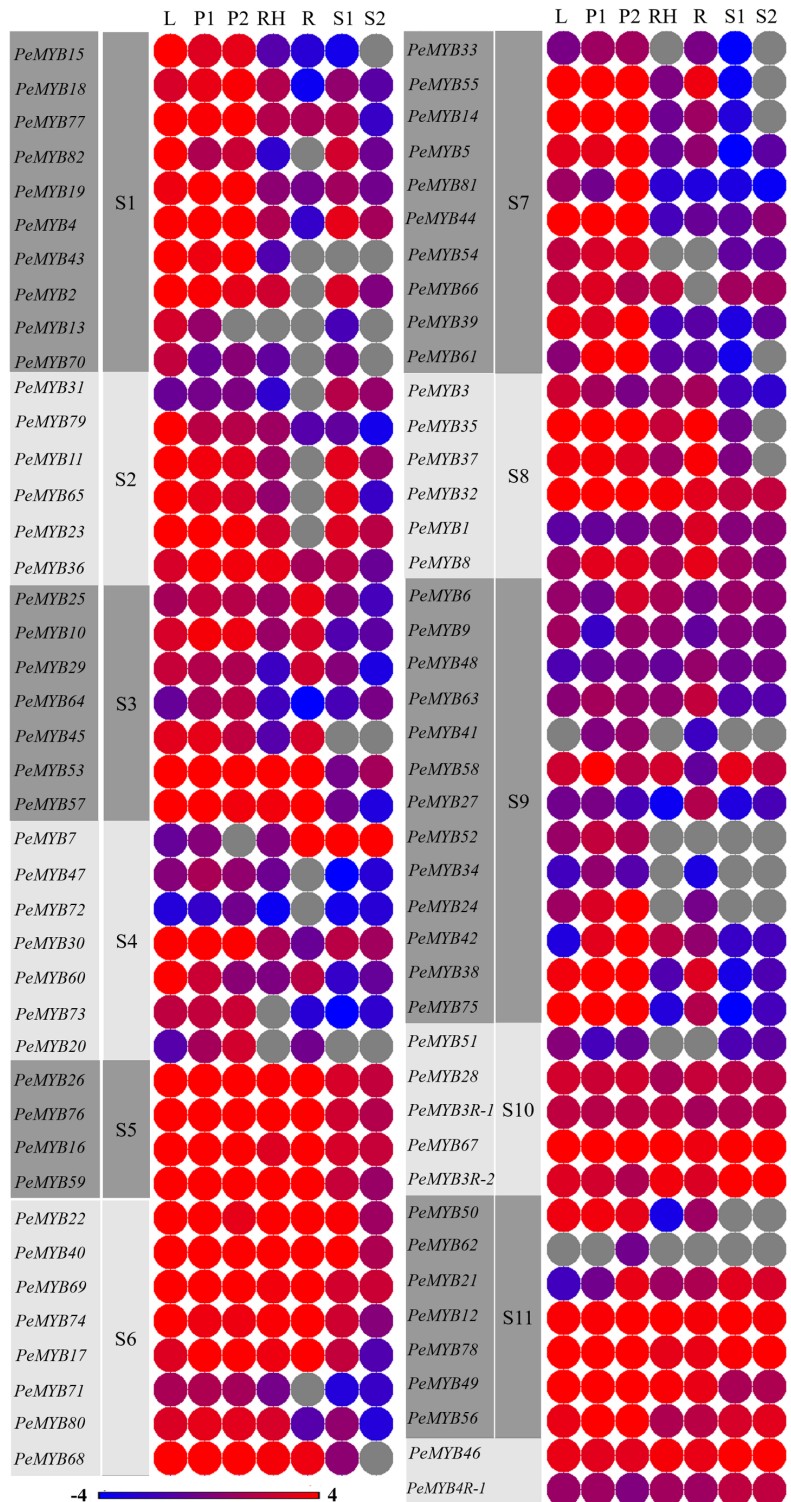

**Figure 4 Expression profiles of *PeMYB*s in different tissues and development stages.** Heatmap showing the expression of 85 *PeMYB*s in different tissues analyzed. Color scale at the bottom of the picture represents log₂ expression values: blue indicating low level and red indicating high level of transcript abundance. L, leaves; P1, early panicles; P2, advanced panicles; R, roots; Rh, rhizomes; SH1, 0.2 m shoots; SH2, 0.5 m shoots.

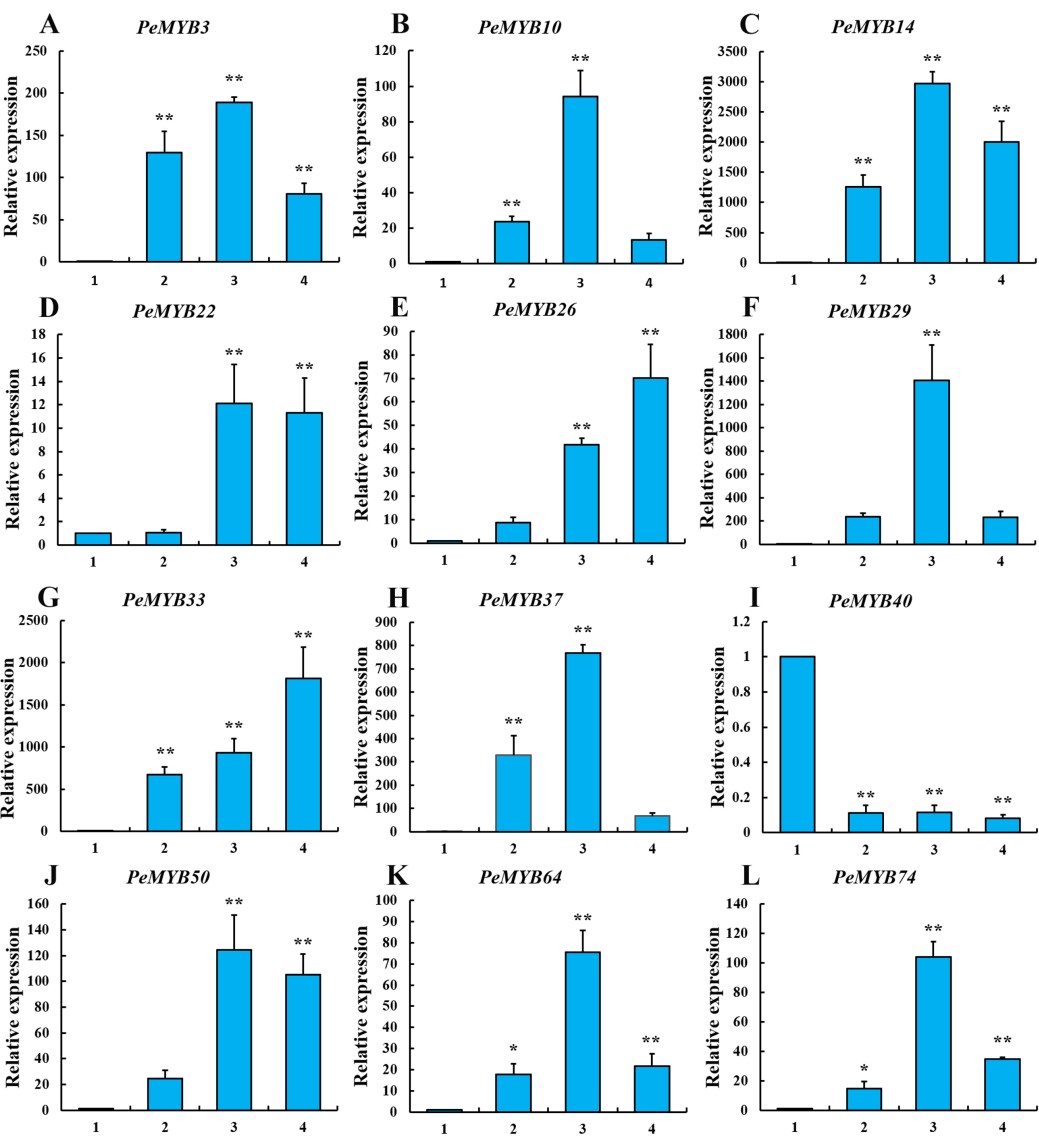

**Figure 5 Expression analysis of 12 *PeMYBs* using qRT-PCR.** *PeTIP41* was used as the reference gene. Average and error bars represent standard deviation of three biological replicates. Asterisks indicate a significant difference between the higher shoots and the 0.2 m shoots ($*p < 0.05$, $**p < 0.01$). (A) *PeMYB3*, (B) *PeMYB10*, (C) *PeMYB14*, (D) *PeMYB22*, (E) *PeMYB26*, (F) *PeMYB29*, (G) *PeMYB33*, (H) *PeMYB37*, (I) *PeMYB40*, (J) *PeMYB50*, (K) *PeMYB64* and (L) *PeMYB74*. 1: 0.2 m shoots; 2: 1.0 m shoots; 3: 3.0 m shoots; 4: 6.7 m shoots.

seven times. The relative expression levels of *PeMYB3*, *PeMYB14*, *PeMYB29*, *PeMYB37*, *PeMYB50* and *PeMYB74* in 3.0 m shoots were upregulated more than 100 times of those in 0.2 m shoots, especially for that of *PeMYB14* and *PeMYB29* up to 2,976 and 1,407 times, respectively. Interestingly, there showed three expression decline patterns in 6.7 m shoots compared to that in the 3.0 m shoots, that is, the expression of *PeMYB10*, *PeMYB29*, *PeMYB37* and *PeMYB64* showed an extremely significant decrease of more than 70%, while that of *PeMYB3* and *PeMYB74* was roughly reduced by half and the expression of *PeMYB22* and *PeMYB50* decreased by approximately 10%.

In general, the lignification degree increased with the growth of bamboo shoots, which was confirmed by the histological slides of different height shoots (Fig. S3). Thus, the higher the bamboo shoots, the higher the degree of lignification. These qRT-PCR results indicated that *PeMYB*s are differentially expressed in bamboo shoots of different height, suggesting that these proteins play different important roles in the lignification of the SCW in moso bamboo. Moreover, the expression of all the *PeMYB*s was verified by using *PeNTB* as a second reference gene, which showed similar results (Fig. S4) and strengthened the reliability of the data generated by using *TIP41*.

## DISCUSSION

The MYB family, as one of the largest and most significant TF families in plants, is involved in regulating various process of plant growth and development and most importantly, the transcriptional regulation of SCW deposition (*Yang & Wang, 2016*; *Wei et al., 2017*; *Sun et al., 2017*). MYB TFs play a vital role in the improvement of material properties and the accumulation of lignocellulosic biomass (*Zhong, Richardson & Ye, 2007*; *Zhong et al., 2008*; *Zhou et al., 2009*). However, little is known about the MYB TFs in bamboo, and less is known about their specific functions in the formation and deposition of SCW. Therefore, the present study focused on the analyses of the gene structure, evolutionary relationship, tissue-specific expression and function prediction of *PeMYB*s, which provided the basis for the further study and practical application of *PeMYB*s.

### Homology of gene structure indicated close evolutionary relationships and similar functions

In the present study, 85 PeMYBs were identified in moso bamboo, and the number of PeMYB members was slightly less than that found in other monocotyledonous and dicotyledonous plants (*Dubos et al., 2010*; *Li et al., 2016c*; *Zhang et al., 2016*). It is likely that the bamboo genome is a draft, which does not cover the entire genome (*Peng et al., 2013a*). In the present study, the members of the MYB gene family were divided into the same subgroups with mostly similar exon/intron structures (Fig. 2). Interestingly, the exons of most of the members in a same subgroup were relatively conserved. For example, the first or first two exons had the same size and position but showed differences in intron length and the position of the last exon, which resulted in a shift in the intron splicing position. This finding suggested that these members were evolutionarily close and might share a similar function. Similarly, previous studies suggested that homologous MYB proteins that cluster together in a subgroup or clade share similar or the same evolutionary origins, which is particularly reflected in the gene structure of the number and size of exons and introns as well as the insertion position of introns (*Dubos et al., 2010*; *Li et al., 2016c*). However, there were also some exceptions with different gene structures from other *PeMYB*s in the same subgroup, which indicated that there might be differences in the evolution of the members in the MYB family and these proteins might have new functions. This finding further supported the diversity of MYB function.
## The variation of amino acids in the conserved DNA-binding domain might change the function and activity

In general, MYB proteins possessed the characteristic of a highly conserved DNA-binding domain in the N-terminus, and the second half of each R structure was particularly conserved (*Kranz, Scholz & Weisshaar, 2000*; *Jiang, Gu & Peterson, 2004*; *Dubos et al., 2010*; *Du et al., 2012b*). In the present study, the third helical structures of PeMYBs were more conserved than the other two helical structures, consistent with the findings of previous studies (*Jiang, Gu & Peterson, 2004*; *Du et al., 2012b*). We speculated that the amino acid sequence of the third helix was particularly important for the function of PeMYBs. On the one hand, the highly conserved amino acids in the third helix may reflect the functional stability of MYBs throughout the long evolutionary process in plant. Moreover, further analysis of a typical characteristic structure from other model pants (*Arabidopsis* and rice) indicated that threonine (T)-88 in the R3 repeats were completely conserved in 82 PeR2R3-MYBs, while the change from threonine (T)-88 to S or tyrosine (Y) was observed in three AtMYBs and seven OsMYBs. This reflected the functional divergence among different plant species (Fig. 1; Figs. S1 and S2).

On the other hand, the species-specific function of PeMYBs may result from the variation of the key amino acids in this region. In addition, the replacement of amino acids within the linker region between R2 and R3 repeats was found in PeMYBs. For instance, the proline (P)-52 in the linker region was substituted by serine (S), which may decrease the stability of the protein-DNA complex and even lead to the loss of DNA binding activity (*Dias et al., 2003*; *Heine, Hernandez & Grotewold, 2004*). The same phenomenon was found in a previous study, and the replacement rate and replacement location were similar (*Li et al., 2016c*). Furthermore, the proline (P) (located on 52) was replaced not only by S in the present study but also by alanine (A) and threonine (T) substitutions (in PeMYB30, PeMYB60 and PeMYB73). Interestingly, compared to previous studies (*Du et al., 2012b*; *Stracke et al., 2014*), more amino acid substitution sites were found in PeMYBs, which was helpful to further study on the evolutionary relationship and function of species-specific MYB TFs in moso bamboo. Considering that amino acid sequence variations may change protein function, we assumed that the species-specific PeMYBs might have new functions, which needs further research to dissect in moso bamboo.

## The expression of *PeMYB*s was tissue-specific and closely related with the development of shoots and panicles

The expression profiles of *PeMYB*s in different tissues and development stages were analyzed and described in detail, which could contribute to further study on the tissue specificity and the dynamic variation rule of *PeMYB*s in moso bamboo. The present study showed some *PeMYB*s, such as the members in S5 and S6, which showed high expression levels in all the detected tissues, suggesting that these genes might play important roles in the growth and development of bamboo by regulating the morphogenesis of various organs. In contrast, some *PeMYB*s, such as the members in S9, were expressed at low levels in all the tissues analyzed, suggesting that these genes likely have unknown

functions in other tissues. This finding was consistent with that of gene structure, which further suggested that structure determines function. Genes with similar structures had similar functions and participated in the same physiological–biological processes or stages of growth and development, and their expression patterns were consistent. Interestingly, expression profiling showed that some *PeMYBs* were not detected in the young bamboo shoots, suggesting that these *PeMYBs* might mainly be involved in the synthesis of lignin and the response to environmental factors. Bamboo shoots with heights of 20 and 50 cm were in the early stages of shoot development, which was mainly in the differentiation and elongation of cells, rather than in the process of lignification.

## The function diversity and universality of PeMYBs

Although rice is phylogenetically near to moso bamboo in the phylogenetic tree (Fig. S5), most of those MYB TFs in rice have not been verified by experiment. Nevertheless, the function of *Arabidopsis* MYB TFs has been well studied and experimentally verified; thus, we predicted the function of moso bamboo MYB TFs based on the well-studied MYBs in *Arabidopsis*. The diverse gene structure implied a diverse gene functions. The MYB TFs of moso bamboo and *Arabidopsis* were divided into 36 functional clades. According to the homology of AtMYBs, the functions of different MYB members largely varied, even if these MYBs were from the same clade. For example, AtMYB4, AtMYB7 and AtMYB32 belonged to C7, and not only regulated SCW synthesis (*Fornalé et al., 2010*; *Shen et al., 2012*; *Hussey et al., 2013*; *Yang & Wang, 2016*) but also were involved in regulating anthocyanin biology and flower development (*Jin et al., 2000*; *Preston et al., 2004*; *Vimolmangkang et al., 2013*; *Fornalé et al., 2014*). The PeMYBs in C4 tended to cluster together with AtMYB16, AtMYB17 and AtMYB106, indicating that these MYBs might be related to the formation of controlling petal epidermal cell morphology and regulating early development of inflorescence (*Jakoby et al., 2008*; *Zhang et al., 2009*) and played important roles in the formation of the trichomes (*Baumann et al., 2007*). In addition, MYBs with the similar functions were scattered in different clades rather than aggregated in the same clade. For example, the PeMYBs involved in the formation of SCW and lignin synthesis were scattered in C1, C12, C14, C16, C22 and C32, indicating that these MYBs might play important and central roles in the SCW synthesis pathway. Importantly, nine clades did not have functional annotation due to a lack of studies in *Arabidopsis* or the species-specific clade of moso bamboo, which required further studies.

## The SCW-related *PeMYBs* showed different patterns with increasing shoot height

To further validate the reliability of the prediction, 12 *PeMYBs* related to SCW synthesis were selected to examine the changes of their transcript levels in the base region of bamboo shoots of four heights at different developmental stages by using qRT-PCR. The results showed that the transcript levels of 11 *PeMYBs* were significantly upregulated with increasing bamboo shoot height, suggesting that they might be positive regulators for the formation of SCW, which is consistent with previous studies of model plants (*Zhong, Richardson & Ye, 2007*; *Zhong et al., 2008*; *Zhou et al., 2009*). On the contrary, the

trend of *PeMYB40* kept at a relatively stable low level after a sharp drop, indicated it might be involved in the negative regulation of SCW formation. In addition, nine *PeMYB*s demonstrated decreased expression in 6.7 m shoots compared to those in 3.0 m shoots, suggesting that these genes likely played more important roles in subsequent physiological and biochemical reactions at earlier developmental stages in bamboo shoots.

Furthermore, most members in same subgroup had similar expression patterns. Exceptionally, *PeMYB2*, *PeMYB40* and *PeMYB74* belonged to C16 showed entirely different expression patterns, suggesting that they might play different roles in the process of SCW lignification at different development stages of bamboo shoots. These results strongly supported the above functional prediction analysis for PeMYBs. However, the functional predictions performed in the present study might not be completely accurate because of the diversity of MYB TF functions between monocotyledons and dicotyledons.

Moreover, MYBs are also associated with biotic and abiotic stresses (*Ashrafi-Dehkordi et al., 2018*), such as powdery mildew (*Li et al., 2016a*), drought (*Shi et al., 2018*) and salt (*Fang et al., 2018*) and heavy metal stresses (*Van De Mortel et al., 2008*; *Yuan et al., 2018*). The regulation of PeMYBs for the growth and development of moso bamboo might involve a complex regulatory network, and more studies are needed.

# CONCLUSIONS

MYB TFs are widely distributed among higher plants and play critical roles in plant grow and development as well as in response to biotic and abiotic stresses. To reveal the status of MYBs in moso bamboo, a genome-wide screening was conducted and a total of 85 PeMYBs were identified, including 82 typical R2R3-MYB proteins, two R1R2R3-MYB proteins and one 4R-like MYB protein, which were classified into 11 subgroups (S1–S11) on the basis of their phylogenetic relationships. Analysis of intron/exon structures indicated that the splicing sites and lengths of most introns were highly conserved in the MYB domains, especially in those within the same subgroups. Based on the phylogenetic relationships and comparing with those well studied MYBs of *Arabidopsis*, the function of PeMYBs were predicted. Especially those function of 12 PeMYBs related to the biosynthesis and deposition of SCW were validated by qRT-PCR, which demonstrated their transcript abundance levels changed significantly with the increasing degree of lignification in bamboo shoots. The comprehensive analyses provided an overall insight into MYB TFs in moso bamboo and their potential involvement in SCW processes. These results will aid in understanding of and conducting further studies on the molecular mechanism of PeMYBs involved in bamboo wood formation, which is helpful for the development and utilization of bamboo lignocellulosic biomass.

### Funding

This work was funded by the Special Fund for Forest Scientific Research in the Public Welfare from State Forestry Administration of China (No. 201504106) and the

Sub-Project of National Science and Technology Support Plan of the Twelfth Five-Year in China (No. 2015BAD04B0101). The funders had no role in study design, data collection and analysis, decision to publish, or preparation of the manuscript.

## Grant Disclosures

The following grant information was disclosed by the authors:
Special Fund for Forest Scientific Research in the Public Welfare from State Forestry Administration of China: 201504106.
Sub-Project of National Science and Technology Support Plan of the Twelfth Five-Year in China: 2015BAD04B0101.

## Competing Interests

All authors are employed by International Centre for Bamboo and Rattan, and have no competing interests.

## Author Contributions

- Kebin Yang conceived and designed the experiments, performed the experiments, analyzed the data, prepared figures and/or tables, authored or reviewed drafts of the paper, approved the final draft.
- Ying Li contributed reagents/materials/analysis tools, approved the final draft.
- Sining Wang contributed reagents/materials/analysis tools, approved the final draft.
- Xiurong Xu contributed reagents/materials/analysis tools, approved the final draft.
- Huayu Sun contributed reagents/materials/analysis tools, approved the final draft.
- Hansheng Zhao contributed reagents/materials/analysis tools, approved the final draft.
- Xueping Li contributed reagents/materials/analysis tools, approved the final draft.
- Zhimin Gao contributed reagents/materials/analysis tools, authored or reviewed drafts of the paper, approved the final draft.

## Data Availability

RNA-seq data have been deposited at EMBL under accession ERP001341.

## Supplemental Information

Supplemental information for this article can be found online at http://dx.doi.org/10.7717/peerj.6242#supplemental-information.

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
