# Peer review of "Genome-wide identification and expression analysis of the MYB transcription factor in moso bamboo (Phyllostachys edulis)"

_PeerJ, doi:10.7717/peerj.6242_

## Round 0.1 · original submission · Major Revisions

Please respond to the queries of our reviewers. Pay special attention to reviewer #2's 1st point, regarding function attribution of your 18 groups: if possible compare them to those of a plant phylogeneticaly closer to bamboo than Arabidopsis.

Personal comment by the editor:
- Please increase the size of the labels in Fig.5

·

Basic reporting

no comment

Experimental design

For phylogenetics analysis, it was based on Maximum Likelihood rather than NJ.
Quantitative verification of gene expression levels must employ at least three biological replicates.
Please provide the evidence for selecting SCW-related PeMYB genes. In addition, the authors should provide the lignin content data at the corresponding sampling point.

Validity of the findings

In the materials and methods, the authors mentioned the rice sequences were used for phylogenetics analysis, but there was not any result shown.

Additional comments

The results section contains a lot of discussion content.

·

Basic reporting

the authors should provide raw data and update the reference.

Experimental design

The methods should be described with sufficient information to be reproducible by another investigator.

Validity of the findings

no comment

Additional comments

1.A phylogenetic tree was constructed with the full-length of MYB amino acid sequences (85 PeMYBs and 132 AtMYBs) to predict the function of moso bamboo and divided into 18 function groups. The authors further depend on the phylogenetic cluster results identified 12 biosynthesis and deposition of the secondary cell wall related MYB genes, this prediction method is too simple and not enough. We know bamboo belongs to the monocotyledonous, perennial grasses, while Arabidopsis is represented dicotyledenous plant. The authors should choose some genes have clear function (including secondary cell wall related genes from different plant, such as grasses plant ) to construct another phylogenetic tree, and according to the cluster result to identify the gene function. This part needs to be strengthened.
2.Figure 1 is not necessary, because I have seen the same figure in other articles. The authors should providing a multiple sequence alignment result (85 PeMYB proteins), and adding some MYB proteins which have a typical characteristic structure from other model pants.
3.Line 63 should insert the reference.
4.Did you repeat when sampling? The sample is single plant or mixed samples? The authors should write clear in Plant Material part.
5.Line 183, What is the concentration of CDNA?
6.Table 1, why does there present two PI,does it mean different analysis content?
7.From line 400 to 402, the authors indicating that these MYBs might play important and central roles in the SCW synthesis pathway, should further analysis the conserved motif, dose these genes shared the same or specific motifs with other PeMYB genes.
8.From line 255 to 257, the authors say Class I including 6 clades (C1, C12, C14, C16, C22 and C32), was responsible for SCW formation by regulating the biosynthesis and deposition of lignin, cellulose and hemicellulose. Class III, including 5 clades (C7, C17, C20, C25 and C29), played important roles in morphogenesis and organogenesis.
In the part of Validation of PeMYBs by using real-time PCR. 12 PeMYBs, including C1, C7, C12, C14, C16, C22, C32, were selected for further validation. Why does only selected C7 in Class III? C12 including 10 PeMYBs, why does only choose two genes? The authors should choose all predicted SCW related genes to validate. I hope the author provides the raw data for this part.
9.The Figure 5 should re-analyse, why does not show * p < 0.05. the gene expression of peMYB64 is not correct.
10.Table S1 should including reference gene primers. If authors use two reference genes, the result will be more reliable.
11.The authors should provide the sequences (gene, cds and protein).

Reviewer 3 ·

Basic reporting

1. The English is needeی to improve. This manuscript requires major English language copyediting.

2. Literature references are sufficient and an appropriate background is provided.

3. In figure 5 the name of font of genes’ names are not clear except PeMYB40. Please, change all fonts same to PeMYB40.

4. In figure 5, it is better to write “relative expression to PeTIp41” rather than relative expression.

Experimental design

1. To draw phylogenetic tree, why NJ method was used?

2. Why PeTIp41gene was used as reference gene?

Validity of the findings

1. The authors should be clear in the goals of this study. They should also be able to clearly express the novelty and significance of this study in the introduction.

2. The research aims are well-defined and are relevant. In this study, the authors use meta-analytic statistical techniques and bioinformatics tools to identify and characterize genes in tomato that are differentially expressed during responses to stress conditions. The analysis is based on publically available microarray tomato gene expression data. The analysis meets the appropriate level of rigor and the results are laid out in sufficient detail.

3. The Conclusion section is minimal, and the authors do not provide a discussion about how their findings connect with the study aims and how their work fits into the larger framework of the related literature. The manuscripts reads too much like a report.

Additional comments

It was explain above.

---

## Round 0.2 · Minor Revisions

Please correct the remaining minor points. Remove the ambiguity highlighted in reviewer#2's second point.

·

Basic reporting

no comment

Experimental design

no comment

Validity of the findings

no comment

Additional comments

no comment

·

Basic reporting

no comment

Experimental design

no comment

Validity of the findings

no comment

Additional comments

1. It is difficult for me to find author’s change, should present the specific line number where have been revised.

2. The authors indicating that these MYBs might play important and central roles in the SCW synthesis pathway, should further analysis the conserved motif, dose these genes shared the same or specific motifs with other PeMYB genes. The author’s answer is “We have performed further analysis of the conserved motif, but we did not find that they shared the same or specific motifs with other PeMYB genes”. There are no same or specific motifs with other PeMYB genes, I think the answer is contradictory.

3. Arabidopsis should be italic.

4. It will be great if homologous gene related with SCW in model plants or reported researches could be discussed here and comparing with the results of this study.

Reviewer 3 ·

Basic reporting

1. There are three references with the first author Li in 2016, they mud be determined by letter a, b and c.
2. The reference “Lipsick JS. 1996. One billion years of MYB. Oncogene 13(2):223-235.” Must move before “Liu CW, Fukumoto T, Matsumoto T, Gena P, Frascaria D, Kaneko T, Katsuhara M, Zhong SH, Sun XL, Zhu YM, Iwasaki I, Ding XD, Calamita G, Kitagawa Y. 2013. Aquaporin OsPIP1;1 promotes rice salt resistance and seed germination. Plant Physiology and Biochemistry 63(63C):151-158 DOI: 10.1016/j.plaphy.2012.11.018.”
3. The reference “Wilkins O, Nahal H, Foong J, Provart NJ, Campbell MM. 2009. Expansion and diversificationof the Populus R2R3-MYB family of transcription factors. Plant Physiology 149(2):981-993 DOI: 10.1104/pp.108.132795.” Must move before “Windham GL, Williams WP, Mylroie JE, Reid CX, Womack ED. 2018. A histological study of aspergillus flavus colonization of wound inoculated maize kernels of resistant and susceptible maize hybrids in the field. Frontiers in Microbiology 9:799 DOI: 10.3389/fmicb.2018.00799.”
4. The reference “Zhou J, Lee C, Zhong R, Ye ZH. 2009. MYB58 and MYB63 are transcriptional activators of the lignin biosynthetic pathway during secondary cell wall formation in Arabidopsis. The Plant Cell 21(1):248-266 DOI: 10.1105/tpc.108.063321.” must be the last reference.
5. The English language is acceptable now.

Experimental design

1.The NJ method is a common method and it is not necessary use maximum likelihood.
2.The replication numbers of experiment is acceptable.
3. methods described well in revised manuscript.

Validity of the findings

1. The result and conclusion sections was revised and is acceptable now.

Additional comments

References must be listed in alphabetical order

---

## Round 0.3 · accepted · Accept

I am satisfied with the introduced changes.

#